# Perioperative Systemic Chemotherapy for Colorectal Liver Metastasis: Recent Updates

**DOI:** 10.3390/cancers13184590

**Published:** 2021-09-13

**Authors:** Hee Yeon Lee, In Sook Woo

**Affiliations:** Department of Internal Medicine, Yeouido St. Mary’s Hospital, College of Medicine, The Catholic University of Korea, Seoul 07345, Korea; urloved@catholic.ac.kr

**Keywords:** colorectal neoplasms, liver, metastasis, neoadjuvant therapy, adjuvant chemotherapy

## Abstract

**Simple Summary:**

The development of cytotoxic chemotherapy, targeted agents and immune check point inhibitors has improved survival outcomes and quality of life in patients diagnosed with metastatic colorectal cancer (CRC). Long-term survival and cure are possible in well-selected CRC patients with liver metastases (LM). The criteria for resectable LM and the eligibility of patients should be evaluated at the time of diagnosis or during the clinical course via a multidisciplinary team approach. The advantages of adjuvant chemotherapy after curative resection of LM are uncertain currently. Systemic preoperative chemotherapy may convert unresectable LM to a resectable type. However, the optimal combination of systemic drugs and treatment strategy has yet to be established. This article summarizes recent reports of perioperative systemic treatment for patients with colorectal liver metastases (CLM). This review provides an update for physicians involved in managing patients with CLM.

**Abstract:**

The liver is the most common site of metastases for colorectal cancer. Complete resection in some patients with resectable liver metastases (LM) can lead to long-term survival and cure. Adjuvant systemic chemotherapy after complete resection of LM improves recurrence-free survival; however, the overall survival benefit is not clear. In selected patients, preoperative systemic treatment for metastatic colorectal cancer can convert unresectable to resectable cancer. This review will focus on patient selection, and integration of perioperative and postoperative systemic treatment to surgery in resectable and initially unresectable LM. Additionally, new drugs and biomarkers will be discussed.

## 1. Introduction

Globally, colorectal cancer (CRC) is the third most common cancer and the second leading cause of cancer death [1]. The World Health Organization (WHO) Global Cancer Observatory (GLOBOCAN) reported nearly 1.8 million (10%) new cases and 0.8 million deaths (9%) of CRC in 2018. Colorectal metastases develop in more than half of all patients, and a majority of them carry liver metastases (LM). Reviews of autopsy data involving patients who died from CRC revealed that liver was the only metastatic site in a third of them [2]. In selected patients, colorectal LM (CLM) is curable. The reported 5-year disease-free survival (DFS) rate in patients who underwent resection of primary tumor and LM was about 20% and the 5-year overall survival (OS) rate was 38% [3]. Ridder et al. reported a retrospective case-matched control study of systemic chemotherapy vs. resection of LM for patients with CRC [4]. The patients who underwent surgery for LM showed superior survival (median OS 26.5 months in chemotherapy group vs. 56 months in surgery group, *p* = 0.027), suggesting that the resection of CLM should always be considered for the selected patients. Resectability of LM and indications for treatment should be evaluated at the time of diagnosis or during the clinical course through a multidisciplinary team (MDT) approach because the 5-year survival rates of patients who underwent complete resection of LM compared with those who do not [5]. Thus, it is critical to adopt the best treatment strategy available for patients with CLM. Currently, based on an MDT approach, the resectability of LM depends on the clinical factors including expected function of the remnant liver, number and size of LM, extrahepatic disease, tumor biology and patient factors. Although better outcome, even cure, can be expected by hepatic resection for patients with CLM, recurrence after surgery is worrisome, thus more effective perioperative systemic chemotherapy is required. This review will focus on patient selection, perioperative systemic treatment for resectable and initially unresectable LM, new drugs and biomarkers.

## 2. Patient Selection for Hepatic Resection and Scoring in Colorectal Liver Metastases

In patients with resectable CLM, the resection of primary and metastatic tumors is the current standard of care and achievement of R0 resection is critical [6]. The definition of resectability has been changing over time. Recently three criteria regarding remaining liver function were suggested as follows: preservation of at least two contiguous hepatic segments, adequate blood flow and biliary drainage and >20% remnant liver of total liver volume [6]. Preoperative chemotherapy can eradicate micrometastases, downsize the tumor and delineate tumor biology. Tumor biology including chemosensitivity can facilitate subsequent treatment strategies. Several scoring systems were developed to predict the recurrence and prognosis in CLM [7,8,9]. Previous scoring systems included disease-free interval, the size and number of LM, the staging of tumor and node, the level of carcinoembryonic antigen (CEA), tumor grade, margin status and age [7,8,9]. Fong et al. suggested five clinical criteria as a clinical risk score including nodal involvement, DFS from the primary to LM < 12 months, number of LM > 1, CEA > 200 ng/mL, the largest tumor >5 cm [8]. Favorable outcome was expected in patients with up to two criteria. The current scoring systems incorporate genetic factors such RAS and BRAF, and extrahepatic disease [7,9]. The Comprehensive Evaluation of Relapse Risk (CERR) score stratified patients who received curative-intent hepatic resection into three groups: low-risk (CERR score 0–1), medium-risk (CERR score 2–3) and high-risk (CERR score > 3). The components of CERR score include: KRAS/NRAS/BRAF-mutated tumor (1 point); node-positive primary (1 point); extrahepatic disease (1 point); CEA level > 200 ng/mL or carbohydrate antigen 19-9 (CA19-9) > 200 U/mL (1 point) and modified tumor burden score (calculated by the size, number and extension of LM) ranging between 5 and 11 (1 point) or 12 and over (2 points) [7]. The prognostic scoring system facilitates individualized optimal therapeutic strategy and selection of patients indicated for perioperative systemic treatment. In patients with high recurrence risk on clinical scoring, pre-operative chemotherapy can be an option. Essentially, the decision about resectability and treatment sequencing should be based on an MDT approach.

## 3. Adjuvant Chemotherapy for Resectable Colorectal Liver Metastases

Upfront surgery is a therapeutic option for patients with limited LM. The treatment goal for resectable LM is cure and long-term survival. The definition of resectability remains a challenge and is still evolving, and the reported resection rate is mainly biased toward high-volume centers. The European Colorectal Metastases Treatment Group (ECMTG) proposed that resection should be considered for patients with more than 30% post-surgery even after portal vein embolization, absence of celiac lymph node involvement or extra-hepatic disease, and without invasion of two branches of liver pedicle, inferior vena cava or three hepatic veins [10]. The rationale for adjuvant chemotherapy after resection of primary tumor and LM is based on the efficacy of chemotherapy in adjuvant setting for stage III CRC and palliative settings for metastatic CRC. However, this efficacy is not directly related to adjuvant settings after resection of LM. A few clinical trials were designed to establish the best treatment strategy in resectable LM; however, poor recruitment, the need for relatively long duration of follow-up and the use of old drugs such as oral uracil-tegafur, 5-fluorouracil/leucovorin (5-FU/LV) were limitations [11,12,13,14]. The clinical trials in resectable CLM are summarized in Table 1. The FFCD ACHBTH AURC 9002 trial compared surgery including resection of primary tumor and LM with surgery followed by 5-FU/LV chemotherapy [13]. The trial was based on 171 patients. The 5-year DFS rate was 33.5% in patients undergoing surgery followed by chemotherapy (5-FU/LV), 26.7% in those treated with surgery alone (*p* = 0.028), and the 5-year OS rate was 51.1% vs. 41.1% (*p* = 0.13). Post-operative 5-FU/LV improved DFS but not OS. Surgery followed by 5-FU/LV and irinotecan (FOLFIRI) in the adjuvant treatment was compared with surgery followed by 5-FU/LV [14]. A total 321 patients were enrolled and the median DFS was 24.7 months for FOLFIRI vs. 21.6 months in 5-FU/LV (HR 0.89, *p* = 0.44), and the 3-year OS was 72.7% vs. 71.6% (HR 0.75, *p* = 0.44). FOLFIRI in the adjuvant setting showed no benefit compared with 5-FU/LV in terms of DFS and OS. However, FOLFIRI within 42 days of resection for LM showed a tendency for improved DFS (HR 0.75, *p* = 0.17), while no difference was shown in patients who received chemotherapy > 42 days after the surgery (HR 1.07, *p* = 0.75). Thus, the importance of early cytotoxic regimen was suggested to eradicate active micrometastases. Sandwiched perioperative chemotherapy is another option. The EORTC 40983 trial reported the efficacy of perioperative chemotherapy with FOLFOX4 and surgery compared with surgery alone, and the primary endpoint was PFS [15]. Patients received six cycles of FOLFOX4 chemotherapy both preoperatively and postoperatively. The trial enrolling 364 patients demonstrated the benefit of perioperative chemotherapy in terms of PFS compared with surgery alone (3-year PFS rate in patients with resection, 42.4% vs. 33.2%; HR 0·773, *p* = 0·025). The OS tended to be prolonged but was not statistically significant between two groups (median OS 63.7 months vs. 55 months, HR 0.84, *p* = 0.3) [16]. Suggested reasons for the lack of difference in OS include limited number of enrolled patients, limitations associated with surgical techniques, and subsequent chemotherapy in cases of recurrence. Based on the PFS demonstrated in this trial, perioperative chemotherapy with FOLFOX remains one of the standard treatments for resectable CLM.

Several meta-analyses of studies investigating systemic chemotherapy in resectable CLM showed the advantage of combined chemotherapy and surgery compared with surgery alone in DFS or PFS [17,18]. However, there was no significant improvement in OS and chemotherapy increased the post-operative complications.

Based on the PFS benefit of the EORTC 40983 trial, the New EPOC trial assessed the role of perioperative chemotherapy with or without cetuximab in resectable CLM with wild-type KRAS [19]. Addition of cetuximab to chemotherapy has shown survival benefits in advanced or inoperable metastatic CRC with wild-type KRAS. Among the 272 patients enrolled, the median PFS was 22.2 months in the chemotherapy group and 15.5 months in the chemotherapy plus cetuximab group (HR 1.17, *p* = 0.304). The median OS was 81 months and 55.4 months, respectively (HR 1.45, *p* = 0.036). Addition of cetuximab to chemotherapy in the perioperative setting resulted in a significant disadvantage in terms of OS [19]. Right-sided CRC was less responsive to EGFR inhibitor treatment in a metastatic setting, and thus the current guidelines recommend EGFR inhibitors for left-sided CRC. The concept of tumor-sidedness was not investigated in the New EPOC trial. Relatively few patients had right-sided CRC in the New EPOC trial, and the impact of sidedness was not significant. Therefore, the role of sidedness in perioperative chemotherapy should be evaluated in future trials. Any RAS-mutated CRC including KRAS and NRAS or BRAF-mutated CRC does not respond to EGFR inhibition, and thus those tests are currently mandatory before anti-EGFR therapy. An additional extended RAS/RAF test was performed using the available samples in the New EPOC trial. In spite of the limited sample size for definite conclusion, the post hoc analysis showed almost identical results in all wild-type RAS/RAF in the whole trial population. Patients receiving chemotherapy and cetuximab showed progression in multiple sites and a high number of early deaths compared with the chemotherapy group. Thus, the role of cetuximab was suggested in accelerating disease progression via development of aggressive phenotype or genotype. There is some criticism of the New EPOC trial [20]. Organizing clinical trials involving both surgery and chemotherapy is complicated, and thus few studies have been completed. There was no validation about the quality of surgery in the New EPOC trial. Several imbalances between two arms were suggested, including resection margin, ablated metastases and resected metastases which can affect the outcome. Therefore, a concern was raised about changing clinical practice according to the results. The BOS2 trial (NCT 01508000) was designed to compare the efficacy of FOLFOX, FOLFOX and bevacizumab, and FOLFOX and panitumumab in patients with wild-type KRAS under perioperative settings. The BOS2 trial was terminated early due to poor recruitment. Nonetheless, evidence does not support adding a biological agent to a cytotoxic doublet to improve the outcome in resectable CLM compared with a cytotoxic doublet alone in a postoperative setting. In resectable CLM, the goal of perioperative chemotherapy is eradication of micrometastases. The results of perioperative chemotherapy and adjuvant chemotherapy appear to be consistent with the study in the adjuvant setting in locally advanced CRC, which showed no benefit of adding biologic agents to a cytotoxic doublet [21,22]. The JCOG0603 trial compared hepatectomy followed by modified fluorouracil, leucovorin and oxaliplatin (mFOLFOX6) with hepatectomy alone in CLM, and preliminary data were presented at the 2020 American Society of Clinical Oncology (ASCO) meeting [23]. The 3-year DFS rate was 52.1% in the mFOLFOX group and 41.5% in the surveillance group (HR 0.63, *p* = 0.002). The 3-year OS rate was 86.6% vs. 92.2% (HR 1.35) and the 5-year OS rate was 69.5% vs. 83%. The median OS after recurrence was 38.4 months vs. 87.6 months. Post-operative mFOLFOX showed superior results in terms of the DFS without improving the OS, which resulted in early termination of the trial. The reasons suggested for the harmful effect of mFOLFOX6 on OS were restricted oxaliplatin use for recurrence and the emergence of more aggressive phenotypes after post-operative chemotherapy. In the JCOG0603 trial, the resection of LM was performed in patients with small size (<5 cm) and limited number (≤3) of LM. The aforementioned trials of post-operative chemotherapy included patients with LM ≥ 5 cm in size or ≥4 in number. [13,14]. In the FFCD ACHBTH AURC 9002 trial, nearly a quarter of the enrolled patients had LM larger than 5 cm. Thus the 5-year OS rate in the group of patients undergoing surgery was higher (83%) in the JCOG0603 trial compared with the FFCD ACHBTH AURC 9002 trial (41%) due to smaller and fewer LM.

Based on all the studies to date, adjuvant chemotherapy after complete resection of CLM or perioperative chemotherapy suggests clinical benefit in PFS or DFS but not in OS. Currently, FOLFOX or capecitabine plus oxaliplatin without targeted agent is considered as an optimal regimen of perioperative systemic chemotherapy for the patients with CLM. Further, one of the challenges for patients with CLM is that oxaliplatin or irinotecan, which are the primary cytotoxic chemotherapeutics in CRC, might cause hepatotoxicity including sinusoidal dilatation and steatohepatitis. Karoui et al. reported that postoperative complications were associated with the number of cycles of preoperative chemotherapy [24,25]. Thus, the recommended duration of perioperative chemotherapy is about 6 months.

## 4. Perioperative Systemic Treatment for Initially Unresectable Colorectal Liver Metastases

Preoperative systemic chemotherapy with biologics for initially unresectable CLM may enable surgery and increase the chances of cure or long-term survival. However, some patients may progress during preoperative chemotherapy and more active systemic control is required considering the patient characteristics. The criteria for initially unresectable CLM are disputed [27]. Table 2 summarizes trials investigating potentially resectable CLM. Gruenberger et al. and Wong et al. evaluated the efficacy and safety of capecitabine, oxaliplatin (CAPOX) and bevacizumab [28,29]. Fifty-six patients with potentially resectable CLM received six cycles of CAPOX and bevacizumab and the 6th cycle did not include bevacizumab [28]. Surgery was performed 5 weeks after administration of last bevacizumab without increased bleeding events or wound-healing complications. Liver regeneration was not affected either. This study suggested that an interval of 5 weeks between bevacizumab administration and surgery is safe. Forty-six patients with poor-risk CLM (number of LM ≥ 4, size of LM ≥ 5cm, unlikely R0 resection and inadequate remnant liver function) received CAPOX plus bevacizumab. The ORR was 78% and 40% of patients converted to resectable [29]. The CELIM study was a phase 2 trial designed to establish the response to and the secondary resectability of neoadjuvant chemotherapy (FOLFOX or FOLFIRI) with cetuximab in patients with unresectable CLM [30]. In the trial, 5 or more than 5 LM was included in the criteria for unresectability. Among the 111 patients enrolled, the response rate was 68% in the group exposed to FOLFOX with cetuximab and 57% in the group treated with FOLFIRI combined with cetuximab (OR 1.62, *p* = 0.23). The KRAS mutation status was analyzed retrospectively, and the results showed a response rate of 70% in patients with wild-type KRAS and 41% in patients with KRAS mutation (OR 3.42, *p* = 0.008). At baseline, the resectability rate was 32% which increased to 60% after chemotherapy (*p* < 0.0001). Additional long-term data of CELIM study was reported [31]. The median PFS was 10.8 months in the group exposed to FOLFOX with cetuximab and 11.2 months in those treated with FOLFIRI combined with cetuximab (HR 1.18, *p* = 0.4). The median OS was 35.8 months vs. 29 months (HR 1.03, *p* = 0.9). The median PFS in patients who achieved R0 resection was 9.9 months and the 5-yesr OS rate was 46.2%. Both regimens of FOLFOX and FOLFIRI containing cetuximab appeared to represent appropriate therapeutic options for initially unresectable CLM in conversion surgery.

The FIRE-3 trial (AIO KRK-0306) evaluated FOLFIRI plus cetuximab or bevacizumab in metastatic CRC and following surgery [32]. Subgroup analysis was carried out among patients with liver-limited disease and wild-type KRAS (*n* = 133) [33]. Based on treatment benefit including objective response rate, early tumor shrinkage (ETS), depth of response (DpR) and OS, FOLFIRI plus cetuximab was preferable over FOLFIRI plus bevacizumab. ETS is defined by tumor shrinkage of ≥20% at 6 weeks of treatment, and DpR refers to the maximum percent change in tumor size compared with baseline [34]. Both ETS and DpR are strongly linked and were identified as prognostic factors in CRC. Additional analysis in FIRE-3 regarding sidedness involved the whole population, without focusing on liver-limited disease. In left-sided CRC, FOLFIRI plus cetuximab resulted in longer PFS compared with FOLFIRI plus bevacizumab (17.6 months vs. 14.1 months, HR 0.65, *p* = 0.002), while the right-sided CRC showed no significant difference (11.0 months vs. 12.4 months, HR 1.02, *p* = 0.94). Patients with left-sided cancer and wild-type KRAS appear to be indicated for FOLFIRI plus cetuximab than FOLFIRI plus bevacizumab. Despite the lack of significant differences in treatment with cetuximab and bevacizumab for right-sided cancer with wild-type KRAS in the FIRE-3 trial, the poor response to anti-EGFR antibody in right-sided cancer is well-known, suggesting that the addition of bevacizumab is reasonable in right-sided colon cancer with wild-type KRAS.

The ATOM trial is the first randomized trial investigating mFOLFOX6 plus bevacizumab versus mFOLFOX6 plus cetuximab for initially unresectable CLM with wild-type KRAS [35]. A total of 122 patients were enrolled, and the criteria for unresectability included the LM number (≥5) and size (≥5 cm). The median PFS was 11.5 months in the bevacizumab group and 14.8 months in the cetuximab group (HR 0.803, *p* = 0.33). The response rate was 68.4% and 84.7%, and the resection rate was 56.1% and 49.2%, respectively. A higher pathological response (grade 1b/2/3) rate was identified in the cetuximab group (92.6%) compared with the bevacizumab group (66.6%, *p* = 0.0229). The median PFS in patients who underwent conversion surgery was 13.8 months in the cetuximab group, and 6.5 months in the bevacizumab group (HR 0.610 [95% CI: 0.298–1.245]) [35]. Although the cetuximab group had a better pathologic response rate and resection rate, the efficacy of cetuximab and bevacizumab was similar terms of in PFS and adverse effects were acceptable in both groups. Fewer LM (1–4) favored cetuximab compared with bevacizumab (HR 0.26, 95% CI 0.084–0.811). Regarding the higher pathologic response of cetuximab and characteristics of size and number of LM favoring cetuximab, the FOLFOX regimen combined with cetuximab represents a better option in patients with fewer but larger LM.

The BECOME trial assessed the efficacy of mFOLFOX plus bevacizumab compared with mFOLFOX in cases of unresectable CRC LM with RAS mutation in a single-center study conducted in China [36]. The primary endpoint was conversion resection rate and secondary outcomes were tumor response, OS, PFS and toxicity. A total of 241 patients were enrolled and the median number of cycles of preoperative bevacizumab treatment was 4 (range, 3–10 cycles). Bevacizumab was discontinued 4–5 weeks before liver surgery. The R0 resection rate was 22.3% in mFOLFOX plus bevacizumab group and 5.8% in mFOLFOX group (*p* < 0.01). The ORR was 54.5% vs. 36.7% (*p* < 0.01), the median PFS was 9.5 vs. 5.6. months (*p* < 0.01) and the median OS was 25.7 vs. 20.5 months (*p* = 0.03). Shorter median OS compared with other clinical trials could be explained by patient characteristic of RAS mutation. Adding bevacizumab improved resectability of LM and long-term survival, but more frequent proteinuria (9.9% vs. 3.3%, *p* = 0.04) and hypertension (8.3% vs. 2.5%, *p* < 0.05) were detected. The OS is also influenced by subsequent treatment after progression, but the study did not determine the specific protocol to investigate further.

The combination of triplet cytotoxic chemotherapy (5-fluorouracil, oxaliplatin and irinotecan: FOLFOXIRI) and biologic agents showed efficacy in unresectable CRC. A meta-analysis of 5 eligible trials (CHARTA, OLIVIA, STEAM, TRIBE and TRIBE2) of FOLFOXIRI and bevacizumab was reported [37]. Treatment with FOLFOXIRI and bevacizumab prolonged PFS (median, 12.2 months vs. 9.9 months; HR 0.74, *p* = 0.001) and OS (median, 28.9 months vs. 24.5 months; HR 0.81, *p* = 0.001) and enhanced the R0 resection rate (16. 4% vs. 11.8%, *p* = 0.007) compared with chemotherapy doublets and bevacizumab. In terms of toxicity, a moderate increase was found in the triplet and bevacizumab group, and patients with BRAF mutation derived no increased benefit. Tumor sidedness is a strong prognostic factor and predictor of the activity of anti-EGFR agents in metastatic CRC [38,39]. Due to the limited benefit of anti-EGFR agents in right-sided CRC, cetuximab or panitumumab is recommended in case of left colon cancer and RAS or wild-type BRAF, especially for conversion. Thus, the combination of FOLFOXIRI and bevacizumab is an option in unresectable CLM with good performance status (Eastern Cooperative Oncology Group performance status 0–1) and right-sided and/or RAS or BRAF mutation. The ongoing TRIPLETE trial compares mFOLFOXIRI plus panitumumab versus mFOLFOX6 plus panitumumab in unresectable CLM with RAS and wild-type BRAF. Response evaluation for resectability should be performed regularly by MDT, given the time to maximal response is 12–16 weeks of systemic treatment. In cases of resectable downsizing, prompt surgery is critical due to the well-known hepatotoxicity (sinusoidal dilatation and steatosis) of oxaliplatin and irinotecan. The benefit of continued systemic treatment after resection of LM is unclear. The role of biologic agents after resection of CLM also remains elusive. In most trials, biologic agents were stopped after surgery and cytotoxic chemotherapy was continued up to a total of 12 cycles. In a retrospective study, the addition of bevacizumab following resection for CLM had no impact on both PFS and OS [40].

## 5. Immune Checkpoint Inhibitors in Colorectal Liver Metastases

In metastatic CRC, checkpoint inhibitors are recommended as an option for patients with deficient mismatch repair (dMMR) or microsatellite instability-high (MSI-H) since a phase 2 study has shown the efficacy of anti-programmed death 1 immune checkpoint inhibitor, pembrolizumab [43]. Of the total 41 patients enrolled, 11 had CRC with dMMR, 21 had CRC with proficient MMR and 9 had non-CRC with dMMR. More than half of enrolled patients with CRC manifested liver metastasis. Objective response evaluation was performed at 12 weeks and every 8 weeks thereafter. Immune-related objective response rate and immune-related PFS rate at 20 weeks were 40% and 78%, respectively, in CRC patients with dMMR, and were 0% and 11%, respectively, in patients with MMR-proficient CRC. The median PFS and OS were not reached in dMMR CRC but were 2.2 and 5.0 months, respectively, in MMR-proficient CRC (PFS, HR 0.1, *p* < 0.001; OS, HR 0.22, *p* = 0.05). Patients with dMMR non-CRC manifested similar responses compared to patients with dMMR CRC, and the median time to response of patients with dMMR CRC was 28 weeks. The CheckMate-142 trial was also conducted to confirm the efficacy of nivolumab in metastatic CRC patients with dMMR or MSI-H [44]. A total of 74 CRC patients were enrolled in the trial and most of them were previously treated. The ORR was 68.9%, and the median time to response was 2.8 months (range 1.4–3.2). Nivolumab was suggested as a new therapeutic option in patients with metastatic CRC with dMMR/MSI-H based on this trial, although the sample size was small and the treatment efficacy against LM was not demonstrated. Based on the results of CheckMate-142, the efficacy of CTLA-4 inhibitor ipilimumab combined with nivolumab was assessed against metastatic CRC with dMMR/MSI-H [45]. The doublet was not compared to the anti-PD-1 monotherapy, but showed high response rates, encouraging survival, and manageable adverse events (ORR 55%; PFS and OS rates at 12 months, 71% and 85%, respectively). Due to the low incidence of dMMR or MSI-H in CRC patients, substantial data to support their use in CLM are currently unavailable. A few case reports showed notable response to checkpoint inhibitors [46,47]. In a patient with Lynch syndrome (MSI-H) and CLM treated with neoadjuvant pembrolizumab, a pathologic complete response (CR) was achieved [46]. Two patients with locally advanced rectal cancer had dMMR, Lynch syndrome and high tumor mutation burden (TMB) [47]. They received neoadjuvant nivolumab, and one of the patients achieved pathologic CR and the other manifested clinical CR. Patients with dMMR/MSI-H, Lynch syndrome or high TMB appear to be good responders to immune checkpoint inhibitors. KEYNOTE-177 trial enrolled 307 chemo-naive mCRC patients with MSI-H or dMMR. Pembrolizumab and chemotherapy (5-FU based chemotherapy with or without bevacizumab or cetuximab) were compared, and pembrolizumab showed longer PFS (16.5 months vs. 8.2 months, HR 0.6, *p* = 0.0002). However, more patients showed progressive disease in the pembrolizumab group than the chemotherapy group (29.4% vs. 12.3%) [48]. The mechanism of resistance is yet unclear. In clinical practice, the risk of early progression of checkpoint inhibitors should be kept in mind.

## 6. Biomarkers and Ongoing Clinical Trials in Colorectal Liver Metastases

Medhavi G et al. reported the impact of neoadjuvant chemotherapy on the tumor microenvironment (TME) in microsatellite stable (MSS) CLM patients at the 2021 Gastrointestinal Cancer Symposium [49]. Neoadjuvant chemotherapy resulted in favorable changes in TME of LM. Leukocytes (CD45) and the density of CD3+ and CD8+ T cells increased, and the high T-cell/macrophage ratios were associated with longer survival. Considering the fact that the majority of CRC tumors are MSS and MMR-proficient, a favorable change of TME after neoadjuvant chemotherapy is notable.

In KRAS WT CRC treated with anti-EGFR therapy, high miR-31-3p expression resulted in inferior outcomes [50]. The miR-31-3p expression was assessed in patients with KRA WT who were enrolled in the New EPOC trial. The median PFS in patients with mid- or high miR-31-3p expression was 12.3 months in the chemotherapy plus cetuximab group and 26.7 months in the chemotherapy group (HR 2.28, *p* = 0.006). In the subgroup study of miR-31-3p expression in operable CLM, patients with low levels of miR-31-3p in the New EPOC trial were not harmed by cetuximab, and miR-31-3p was suggested as a predictive biomarker for the patients with KRAS WT receiving anti-EGFR treatment [51]. TP53, PIK3CA, APC, expression of Ki-67 and MSI with KRAS and BRAF were suggested as meaningful prognostic and predictive markers in CLM [52]. BRAF mutation is an adverse prognostic biomarker in patients undergoing hepatic resection for CLM [53]. In the retrospective study of patients with curatively resected CLM, the incidence of BRAF mutation was 5.5% in CLM, and BRAFV600E mutation was associated with shorter OS and DFS [54]. Analysis of the changing impact of prognostic factors such as clinicopathologic and genetic factors on conditional overall survival over time, BRAF mutation was the worse prognostic factor in the first year after hepatectomy for CLM, whereas surgical margin status and resection of extrahepatic disease were important thereafter. Thus, in cases of recurrence with extrahepatic disease after resection for CLM, additional resection is an option [55]. Kawaguchi et al. reported that RAS, TP53 and SMAD4 are superior to RAS alone in predicting prognosis in CLM [56]. Multigene testing was done in 507 patients who underwent CLM resection. Frequently mutated genes were TP53 (70.8%), APC (53.5%), RAS (50.7%), PIK3CA (15.8%) and SMAD4 (11.0%). BRAF was mutated in 2.0% of patients. Mutations in BRAF, RAS, TP53 and SMAD4 were associated with survival, and coexisting mutations in RAS, TP53 and SMAD4 showed worse survival compared with single or double mutations. This study suggested that RAS mutation status alone is not enough to predict prognosis accurately in patients with CLM. R-spondins are known as oncogenic drivers in CRC. R-spondin/Wnt axis is associated with vascular endothelial growth factor-dependent angiogenesis. In the analysis, 773 patients from FIRE-3 and TRIBE trials who received FOIFIRI/bevacizumab or FOIFIRI/cetuximab were included. About one third (250) of patients carried LM. RAS wild-type patients with any G allele of the R-spondin 2 gene (RSPO2) rs555008 single nucleotide polymorphism (SNP) manifested longer OS compared with those carrying TT genotype (29.0 vs. 23.6 months, *p* = 0.009) [57]. In contrast, any G allele carriers with RAS mutant CRC resulted in shorter PFS compared with TT genotype (8.1 vs. 11.2 months, *p* = 0.023). Genotyping of the RSPO2 rs555008 SNP may facilitate selection of patients who stand to gain mostly from the addition of bevacizumab to FOLFIRI [35]. The impact of SNP within the R-spondin1-3 genes on the results of perioperative chemotherapy or the clinical outcome in the subgroup with LM only was not demonstrated in the study. The biomarkers are summarized in Table 3.

Grouping based on gene expression has been used in many types of cancers. Guinney et al. suggested four consensus molecular subgroups (CMS), a transcriptome-based classification, as follows: CMS 1 (microsatellite instability immune), CMS 2 (canonical), CMS 3 (metabolic) and CMS 4 (mesenchymal) [58]. CMS classification yields deeper insight into the biology of CRC and is of prognostic value in metastatic CRC. However, its clinical applications in treatment have yet to be defined. Stintzing et al. grouped patients who were enrolled in the FIRE-3 trial according to CMS [59]. In CMS3 and CMS4, OS was longer in FOLFIRI/cetuximab group than in FOLFIRI/bevacizumab group. Mooi et al. used CMS classification of patients in the AGITG MAX trial to establish the predictive effect of bevacizumab [60]. CMS2 and CMS3 preferred bevacizumab over other CMS groups. For the clinical implication of CMS, further validation is required in CMS-grouped patients and an umbrella trial design seems to be appropriate. This CMS could be also considered to plan perioperative systemic treatment for patients with CLM in the future.

A few studies investigating combined local treatment including internal radiation with Yttrium, ablative therapy, and hepatic artery infusional chemotherapy (HAIC) has been reported. The results of those combined local treatments are currently debated [55,56]. Clinical trials are currently investigating the combination of HAIC with systemic treatment and the primary objective involves determination of the conversion rate to complete resection in patients with CLM (Table 4).

## 7. Conclusions

In patients diagnosed with CLM, the design of a treatment strategy via an MDT approach is essential given the clinicopathological, genetic and patient factors. Therapeutic options to improve complete resection rates and outcomes of patients include perioperative chemotherapy, pre-and post-surgery and surgery followed by adjuvant chemotherapy. In patients with resectable CLM, upfront surgery followed by chemotherapy can be an option. In patients with resectable but high risk CLM, neoadjuvant chemotherapy may be considered to test tumor biology and to avoid ineffective surgery. For unresectable CLM, good responders to chemotherapy can undergo conversion surgery. The optimal agent associated with a high resection rate, elucidation of the sequence of treatments and the role of biologic agents after hepatic resection in patients with CLM are still unclear. Incorporation of immune checkpoint inhibitors or new targeted agents into clinical trials of perioperative systemic treatment for CLM might improve resectability and clinical outcome. Accurate patient stratification using recently validated biomarkers and detailed clinical trial design including sidedness, genetic factors and treatment sequencing are also required to achieve better outcomes for patients with CLM.

## Figures and Tables

**Table 1 cancers-13-04590-t001:** Clinical trials investigating resectable colorectal cancer with liver metastases.

Trial (Year)	No. of Patients (Treatment vs. Control)	Treatment	Control	DFS or PFS (Median)	OS (Median)
FFCD ACHBTH AURC 9002 Trial (2006) [13]	173 (86 vs. 87)	Sugery followed by 5-FU/LV	Surgery	5-yr DFS rate 33.5 vs. 26.7% (*p* = 0.028)	5-yr OS rate 51.1 vs. 41.1% (*p* = 0.13)
Ychou et al. (2009) [14]	321 (161 vs. 160)	Surgery followed by FOLFIRI	Surgery followed by 5-FU/LV	24.7 vs. 21.6 M (*p* = 0.44)	3-yr OS rate 72.7 vs. 71.6% (*p* = 0.69)
EORTC 40983 (2013) [16]	304 (152 vs. 152)	Surgery + perioperative chemotherapy (FOLFOX)	Surgery	3-yr DFS rate 36.2 vs. 28.1% (HR 0.77, *p* = 0.041)	63.7 vs. 55 M (HR 0.84, *p* = 0.3)
New EPOC (2020) [19]	257 (129 vs. 128)	Surgery + perioperative chemotherapy + cetuximab	Surgery + peroiperative chemotherapy	15.5 v. 22.2 M (HR 1.17, *p* = 0.304)	55.4 vs. 81 M (HR 1.45, *p* = 0.036)
JCOG0603 (2020) [23,26]	300 (151 vs. 149)	Surgery followed by mFOLFOX	Surgery	3-yr DFS rate 52.1 vs. 41.5% (HR 0.63, *p* = 0.002)	5-yr OS rate 69.5 vs. 83%

Abbreviations: DFS: Disease-free survival; PFS: Progression-free survival; OS: Overall survival; M: months; 5-FU/LV: 5-fluorouracil and leucovorin; FOLFIRI: 5-FU/LV and irinotecan; FOLFOX: 5-FU/LV and oxaliplatin.

**Table 2 cancers-13-04590-t002:** Clinical trials investigating initially unresectable colorectal cancer with liver metastases.

Trial (Year)	No. of Patients (Treatment vs. Control)	Treatment	Control	DFS or PFS (Median)	OS (Median)	ORR/ Resection Rate
Gruenberger et al. (2008) [28]	56	CAPOX + Bevacizumab	NA	NA	NA	ORR 73.2% Resection rate 92.9%
Wong et al. (2011) [29]	46	CAPOX + Bevacizumab	NA	12 M PFS rate 50%	12 M OS rate 86%	ORR 78% Resection rate 40%
CELIM (2010, 2014) [30,31]	111 (56 vs. 55)	mFOLFOX6 + Cetuximab	mFOLFIRI + Cetuximab	11.2 vs. 10.5 M (HR 1.18, *p* = 0.4)	35.8 vs. 29 M (HR 1.03, *p* = 0.9)	68 vs. 57% (OR 1.62 *p* = 0.23)/ R0 rate 38 v. 30%
FIRE-3 * (2018) [33]	120 (62 vs. 58)	FOLFIRI + Cetuximab	FOLFIRI + Bevacizumab	11.2 vs. 12.4 M (*p* = 0.74)	40 vs. 33.3 M (*p* = 0.84)	ORR 79 vs. 72.4% (*p* = 0.52)
ATOM (2019) [35]	122 (61 vs. 61)	mFOLFOX6 + bevacizumab	mFOLFOX + Cetuximab	11.5 v. 14.8 M (HR 0.803, *p* = 0.33)	30.4 M vs. NR (HR 0.827, *p* = 0.56)	68.4 vs. 84.7% (*p*-0.0483)/56.4 vs. 49.2%
BECOME trial (2020) [36]	241 (121 vs. 120)	FOLFOX + Bevacizumab	FOLFOX	9.5 vs. 5.6 M (*p* < 0.01)	25.7 vs. 20.5 M (*p* = 0.03)	54.5 vs. 36.7% (*p* < 0.01)/ 22.3 vs. 5.8% (*p* < 0.01)
TRIBE (2015) [41]	508 (252 vs. 256)	FOLFOXIRI + Bevacizumab	FOLFIRI + Bevacizumab	12.3 vs. 9.7 M (HR 0.77, *p* = 0.006)	29.8 vs. 25.8 M (HR 0.8, *p* = 0.03)	ORR 65 vs. 54% (OR 1.56, *p* = 0.013)
TRIBE2 (2020) [42]	679 (339 vs. 340)	FOLFOXIRI + Bevacizumab	FOLFOX + Bevacizumab	19.2 vs. 16.4 M (HR 0.74, *p* = 0.0005)	27.3 vs. 22.5 M (HR 0.82, *p* = 0.032)	ORR 62 vs. 50% (OR 1.61, *p* = 0.0023)

* Population of liver-limited disease in FIRE-3. Abbreviations: DFS: Disease-free survival; PFS: Progression-free survival; OS: Overall survival; ORR: objective response rate; CAPOX: capecitabine and oxaliplatin; FOLFIRI: 5-FU/LV and irinotecan; FOLFOX: 5-FU/LV and oxaliplatin.

**Table 3 cancers-13-04590-t003:** Biomarkers in colorectal liver metastases.

Biomarker	Results	Clinical Application/Future Direction
T-cell/macrophage ratio [49]	High T-cell/macrophage ratio in TME after neoadjuvant chemotherapy, longer survival	Selection of patients for immune checkpoint inhibitors after neoadjuvant chemotherapy
miR-31-3p [50,51]	High expression in KRAS WT treated with anti-EGFR therapy, inferior outcome	Predictive biomarker for the patients with KRAS WT receiving anti-EGFR treatment
BRAF [55]	Worse prognostic factor in the first year after hepatectomy for CLM than margin status	Selection of patients for additional resection in cases with recurrence after resection for CLM
TP53, and SMAD4 [56]	Coexisting mutations in RAS, TP53, and SMAD4, worse survival compared with single or double mutations	Accurate prediction of prognosis in patients with CLM
R-spondin [57]	Association with vascular endothelial growth factor-dependent angiogenesis	Selection of patients who gain mostly from the addition of bevacizumab to chemotherapy
CMS [58,59,60]	CMS2 and CMS3 preferred bevacizumab over other CMS groups	Planning perioperative systemic treatment for patients with CLM

Abbreviations: TME: Tumor microenvironment; CMS: consensus molecular subtypes.

**Table 4 cancers-13-04590-t004:** Ongoing clinical trials of perioperative systemic treatment for colorectal liver metastases.

Trial	Patients	Design	Primary Objective
NCT04003792	Unresectable	FOLFIRI + Bevacizumab + HAIC (oxaliplatin)	Conversion rate to complete resection
NCT02102789	Unresectable	FOLFOX + HAIC (Floxuridine and Dexamethasone) vs. FOLFOX	Conversion rate to complete resection
NCT00482222	Resectable	FOLFOX for 12 weeks—Surgery -FOLFOX for 12 weeks vs. FOLFOX + Cetuximab for 12 weeks-Surgery -FOLFOX + Cetuximab for 12 weeks	PFS
NCT00492999	Unresectable	FOLFOX/FOLFIRI +HAIC (Floxuridine and Dexamethasone)	Conversion rate to complete resection
NCT03401294	Unresectable	FOLFOXIRI + Bevacizumab	Conversion rate to complete resection
NCT04552093	Resectable	FOLFOX/FOLFIRI + HAIC (Floxuridine) -Surgery	Completion of 2 cycles (feasibility)

Abbreviations: FOLFIRI: 5-FU/LV and irinotecan; FOLFOX: 5-FU/LV and oxaliplatin; FOLFOXIRI: 5-FU/LV, oxaliplatin and irinotecan; HAIC: Hepatic artery infusional chemotherapy; PFS: Progression-free survival; OS: Overall survival.

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
