# Peer review of "Perioperative Systemic Chemotherapy for Colorectal Liver Metastasis: Recent Updates"

_cancers, 2021, doi:10.3390/cancers13184590_

Round 1

Reviewer 1 Report

All concerns from the previous version of the manuscript have been addressed of adapted.

Before submission, carefully check all the new written sentences on incompleness, for  examples line 44: ..... approach because the 5-year survival rates ..... compared to those who do not. 

Reviewer 2 Report

I enjoyed reading this comprehensive and timely review of factors affecting outcomes after surgical resection of colorectal cancer liver metastases.

Reviewer 3 Report

The manuscript has even improved and I do not have any more comments.

This manuscript is a resubmission of an earlier submission. The following is a list of the peer review reports and author responses from that submission.

Round 1

Reviewer 1 Report

The current manuscript by Lee and Woo gives a very nice overview over the actual data concerning perioperative systemic chemotherapy. I do not have any corrections.

Author Response

Thank you for your comment. 

Reviewer 2 Report

I enjoyed reading this manuscript which I found comprehensive and useful. It might be helpful if the authors added more about definitions of resectability?

I think it would also be useful when discussing the New EPOC study (reference 28) to also discuss the criticism of it by Nordlinger et al (JCO 2014, volume 32, pages 241-244.

Author Response

I enjoyed reading this manuscript which I found comprehensive and useful. It might be helpful if the authors added more about definitions of resectability?

Thank you for the comment. The definition was added.

"The definition of resectability has been changing over time. Recently three criteria regarding remaining liver function were suggested as follows: preservation of at least two contiguous hepatic segments, adequate blood flow and biliary drainage, and > 20% remnant liver of total liver volume [6]."

I think it would also be useful when discussing the New EPOC study (reference 28) to also discuss the criticism of it by Nordlinger et al (JCO 2014, volume 32, pages 241-244.

Thank you for the comment and the criticism was added.

"There is some criticism of the New EPOC trial [20]. Organizing clinical trials involving both surgery and chemotherapy is complicated, and thus few studies have been completed. There was no validation about the quality of surgery in the New EPOC trial. Several imbalances between two arms were suggested including resection margin, ablated metastases, and resected metastases which can affect the outcome. Therefore a concern was raised about changing clinical practice according to the results."

Reviewer 3 Report

Perioperative chemotherapy for livermetastases of colorectal cancer is an important subject in the field and, as mentioned by the authors, the advantage of adjuvant treatment is uncertain. In addition, the most optimal treatment strategy has not yet been established and different strategies have been applied within different countries. Although, the subject is relevant to the field, several issues need attention before publication.   

Below, critical points are listed for each paragraph.

Simple Summary and abstract:

-Line 14: perioperative chemotherapy should be replaced for neoadjuvant or induction since postoperative chemotherapy does not contribute to the conversion rate.

- Abstract lacks a proper summary of findings and a conclusion and should be rewritten.

  1. Introduction:

-In the introductions details on a selection of studies are described. To my opinion the introduction should point out the current situation, gaps and clinical needs. This can be addressed in more detail.  Details on the different studies should be pointed out in the tables or discussed further in the discussion section.

  1. Patient selection for hepatic resection and scoring in colorectal liver metastases:

-The CERR scoring system is only mentioned in this review. I wonder why the also commonly used FONG scoring system is not included in such detail or compared with CERR in its clinical use.  

-It is not described how these scoring systems are used in daily practice in the decisions on resection of CLM, and the flaws of these scoring systems for daily clinical practice.

  1. Adjuvant chemotherapy for resectable colorectal liver metastases:

-One of the clinical problems is the definition of resectable (or local treatable). There is a clinical difference in technically resectable/treatable liver metastases or clinically useful resection. This may depend on the number and size of the liver metastases, and on the occurrence (metachronous/synchronous). It is not cleat which group is described here in this part of the manuscript.

  1. perioperative systemic treatment for initially irresectable CLM

-This part has been recently reviewed and published by Bolhuis et al. Eur J of Cancer 2020 (Conversion strategies with combinations of chemotherapy and targeted agents for liver metastases of colorectal cancer). Only the new or ongoing trials are useful to add.

-It would be of additional value to add a proposed  scheme for chemotherapy and livermetastases. For example, based on the current literature you can argue support that < 3 LM <5 cm propose no adjuvant treatment. For >4 and > 5 cm no differences OS, however longer PFS etc.

  1. Immune checkpoint inhibitors in colorectal liver metastases.

-Details on the general response rates of checkpoint inhibitors in the treatment of metastasized CRC seems out of focus in the present manuscript. Since the scope of this review is perioperative systemic treatment of CLM this paragraph should, to my opinion, be focused on the role of immunotherapy in the perioperative treatment of liver metastases. Is this a valuable treatment strategy for induction treatment or perioperative management of CLM.  Can resection of CLM be omitted in case of response? What are unmet clinical questions and needs?

-As mentioned in line 362 the risk of early progression of checkpoint inhibitors should be kept in mind. This can be supported by the results of the checkmate 177 study.

  1. Biomarkers and ongoing clinical trials in colorectal liver metastases

-Besides NCT00492999 investigating the role of doublet chemotherapy and HAIC, also NCT03366155, NCT04552093, NCT02102789 are actively recruiting amongst other studies. Therefore Table 3 seems an incomplete overview of ongoing clinical trials of perioperative systemic treatment of CLM.

-Adding a table with an overview of promising biomarkers and potential clinical applications strengthens the content of this paragraph in the review and is currently lacking.

  1. Conclusion:

-Conclusions are rather weak and lacking concluding remarks on current practice, knowledge gaps and clinical needs.

Throughout the manuscript spelling needs to be carefully checked, There are many spelling mistakes and incomplete sentences (words lacking), some examples:

-line 3 CR, line 15/16 has yet to be established yet, use CRC and colorectal cancer uniformly, CLM/LM etc is not consistently used

-Line 50-51, poor sentence, part is lacking, also line 115-117.

Author Response

Simple Summary and abstract:

-Line 14: perioperative chemotherapy should be replaced for neoadjuvant or induction since postoperative chemotherapy does not contribute to the conversion rate.

Thank you for the comment. “perioperative” was changed to “preoperative”

- Abstract lacks a proper summary of findings and a conclusion and should be rewritten.

Thank you for the comment and the abstract was rewritten.

  1. Introduction:

-In the introductions details on a selection of studies are described. To my opinion the introduction should point out the current situation, gaps and clinical needs. This can be addressed in more detail.  Details on the different studies should be pointed out in the tables or discussed further in the discussion section.

 Thank you for the comment. Details to be discussed in other part were edited.

  1. Patient selection for hepatic resection and scoring in colorectal liver metastases:

-The CERR scoring system is only mentioned in this review. I wonder why the also commonly used FONG scoring system is not included in such detail or compared with CERR in its clinical use.  

Thank you for the comment.

Previous scoring systems including FONG criteria and their characteristics were mentioned (reference #8). Details of Fong criteria was added. Fong criteria was developed 20 years ago and currently genetic factors are important, as well. CERR score incorporated genetic factors. Thus CERR score was mentioned.

“Fong et al. suggested five clinical criteria as a clinical risk score including nodal involvement, DFS from the primary to LM <12 months, number of LM >1, CEA >200 ng/ml, the largest tumor >5 cm. Favorable outcome was expected in patients with up to 2 criteria."

-It is not described how these scoring systems are used in daily practice in the decisions on resection of CLM, and the flaws of these scoring systems for daily clinical practice.

Thank you for the comment, and the use of scoring systems in daily practice was added.

"In patients with high recurrence risk on clinical scoring, pre-operative chemotherapy can be an option."

  1. Adjuvant chemotherapy for resectable colorectal liver metastases:

-One of the clinical problems is the definition of resectable (or local treatable). There is a clinical difference in technically resectable/treatable liver metastases or clinically useful resection. This may depend on the number and size of the liver metastases, and on the occurrence (metachronous/synchronous). It is not cleat which group is described here in this part of the manuscript.

Thank you for the comment. The definition of resectability is still evolving, and usually means “technically resectable” liver metastase. The definition of resectability was mentioned in the manuscript. Treatable LM and clinically useful resection is hard to define but scoring systems can be helpful. Details should be discussed in MDT.   

  1. perioperative systemic treatment for initially irresectable CLM

-This part has been recently reviewed and published by Bolhuis et al. Eur J of Cancer 2020 (Conversion strategies with combinations of chemotherapy and targeted agents for liver metastases of colorectal cancer). Only the new or ongoing trials are useful to add.

 Thank you for the comment. This is a review article, thus I think meaningful trials should be mentioned even they were mentioned in other articles.

-It would be of additional value to add a proposed  scheme for chemotherapy and liver metastases. For example, based on the current literature you can argue support that < 3 LM <5 cm propose no adjuvant treatment. For >4 and > 5 cm no differences OS, however longer PFS etc.

Thank you for the comment. It would be better if the proposed scheme can be presented. However there are many factors to consider besides the size and number. Thus in our article, review of previous studies and emphasizing the role of MDT seem to be appropriate.  

  1. Immune checkpoint inhibitors in colorectal liver metastases.

-Details on the general response rates of checkpoint inhibitors in the treatment of metastasized CRC seems out of focus in the present manuscript. Since the scope of this review is perioperative systemic treatment of CLM this paragraph should, to my opinion, be focused on the role of immunotherapy in the perioperative treatment of liver metastases. Is this a valuable treatment strategy for induction treatment or perioperative management of CLM.  Can resection of CLM be omitted in case of response? What are unmet clinical questions and needs?

Thank you for the comment. The response rate of checkpoint inhibitors may seem out of focus. Yet, there is no data of checkpoint inhibitors in perioperative setting. However, the drug may be meaningful in selected patients. Thus the data was mentioned, and in perioperative setting, the role of checkpoint inhibitors is unclear.

-As mentioned in line 362 the risk of early progression of checkpoint inhibitors should be kept in mind. This can be supported by the results of the checkmate 177 study.

Thank you for the comment. The results of keynote 177 was added.

"KEYNOTE-177 trial enrolled 307 chemo-naive mCRC patients with MSI-H or dMMR. Pembrolizumab and chemotherapy (5-FU based chemotherapy with or without bevacizumab or cetuximab) were compared, and pembrolizumab showed longer PFS (16.5 months vs. 8.2 months, HR 0.6, p = 0.0002). However more patients showed progressive disease in the pembrolizumab group than chemotherapy group (29.4% vs. 12.3%). The mechanism of resistance is yet unclear."

  1. Biomarkers and ongoing clinical trials in colorectal liver metastases

-Besides NCT00492999 investigating the role of doublet chemotherapy and HAIC, also NCT03366155, NCT04552093, NCT02102789 are actively recruiting amongst other studies. Therefore Table 3 seems an incomplete overview of ongoing clinical trials of perioperative systemic treatment of CLM.

Thank you for the comment. The studies were added to the Table 3. However NCT03366155 is not perioperative setting, thus it was not included.

-Adding a table with an overview of promising biomarkers and potential clinical applications strengthens the content of this paragraph in the review and is currently lacking.

 Thank you for the comment. Table 4 was added.

  1. Conclusion:

-Conclusions are rather weak and lacking concluding remarks on current practice, knowledge gaps and clinical needs.

Thank you for the comment. Several sentences about clinical practice were added to conclusion.

Throughout the manuscript spelling needs to be carefully checked, There are many spelling mistakes and incomplete sentences (words lacking), some examples:

-line 3 CR, line 15/16 has yet to be established yet, use CRC and colorectal cancer uniformly, CLM/LM etc is not consistently used

-Line 50-51, poor sentence, part is lacking, also line 115-117.

Thank you for the comment. Spelling and sentence were checked again and fixed.

I greatly appreciate your detailed comments.